# Systematic Review of Prognostic Models Compared to the Mayo Risk Score for Primary Sclerosing Cholangitis

**DOI:** 10.3390/jcm10194476

**Published:** 2021-09-28

**Authors:** Paul A. Schmeltzer, Mark W. Russo

**Affiliations:** Division of Hepatology Atrium Health, Wake Forest University School of Medicine, Charlotte, NC 28204, USA; paul.schmeltzer@atriumhealth.org

**Keywords:** biomarkers, outcomes, liver transplant, cholangiocarcinoma

## Abstract

Background: Primary sclerosing cholangitis (PSC) is a cholestatic liver disease with a variable clinical course that can ultimately lead to end-stage liver disease, cholangiocarcinoma, and the need for liver transplantation. Several prognostic models have been developed to predict clinical outcomes and have been compared to the revised Mayo Risk Score (rMRS). Aim: To conduct a systematic review comparing the rMRS to other non-invasive prognostic tests for PSC. Methods: A systematic review of studies from 2000 to 2020 was performed that compared non-invasive biochemical prognostic models to the rMRS in predicting outcomes in patients with PSC. Results: Thirty-seven studies were identified, of which five studies that collectively included 3230 patients were reviewed. Outcomes included transplant-free survival or composite clinical outcomes. The rMRS was better than the Amsterdam–Oxford model for predicting 1-year transplant-free survival, c-statistics 0.75 and 0.70, respectively. The UK-PSC score outperformed the rMRS for 10-year transplant-free survival, c-statistics 0.85 and 0.69, respectively. An enhanced liver fibrosis score was independently associated with transplant-free survival after adjusting for rMRS. PREsTo predicts 5-year hepatic decompensation with a c-statistic modestly higher than rMRS; 0.90 and 0.85, respectively. Conclusion: Newer prognostic models, including the UK-PSC score and PREsTo, are more accurate at predicting clinical endpoints in PSC compared to the rMRS. Time frames and clinical endpoints are not standard among studies.

## 1. Introduction

Primary sclerosing cholangitis (PSC) is a chronic cholestatic liver disease characterized by biliary inflammation and fibrosis. The pathogenesis remains elusive though genome-wide association studies have suggested that there is a genetic predisposition that may be triggered by environmental exposures [1]. The diagnosis is typically established by cholangiogram, such as magnetic resonance cholangiography (MRC) showing multifocal strictures and dilation of the intra- and/or extrahepatic bile ducts. Because endoscopic retrograde cholangiopancreatography (ERCP) is invasive, it is typically reserved for treatment of dominant strictures and to evaluate for cholangiocarcinoma. Liver biopsy is utilized to exclude small-duct PSC or autoimmune hepatitis overlap.

There is no proven medical therapy and progressive disease can lead to complications including bacterial cholangitis, hepatobiliary malignancies, biliary cirrhosis, hepatic decompensation, and the need for liver transplant. Population-based studies have shown that PSC is a relatively rare disease with a point prevalence of 6 per 100,000 and a median survival to liver transplant or PSC-related death of 13 to 21 years [2]. Because PSC is a heterogeneous disease with a long interval from diagnosis to clinical endpoints, the development of prognostic models would be informative.

A 2016 publication from the International PSC Study Group acknowledged the need for better surrogate endpoints as existing data did not support level 2 validation (i.e., a validated surrogate endpoint) for any biomarkers [3]. More recently, a number of non-invasive prognostic tests for PSC have been studied. These include biomarkers, non-invasive fibrosis tests, and biliary imaging studies. We previously published a systematic review of non-invasive prognostic models for PSC, demonstrating the availability of biochemical models as well as elastography and MRI scores [4]. The aim of this systematic review is to compare the performance characteristics, time horizons, and clinical outcomes of newer biochemical prognostic models with or without clinical data to the revised Mayo Risk Score (rMRS).

## 2. Methods

### 2.1. Search Terms and Study Inclusion

We conducted a systematic review applying the MOOSE guidelines [5] of non-invasive biochemical prognostic models with or without clinical data for PSC from 2000 (the time of publication of the rMRS) to 2020 from the following databases: PUBMED, Ovid, Cochrane, and EMBASE. We (PS, MR) used the keywords: “primary sclerosing cholangitis”, “biomarkers”, “prognosis”, “prognostic score”, “liver transplant”, “cholangiocarcinoma”, “disease progression”, and “survival” to identify relevant articles in English. The references of articles were reviewed. We hypothesized that newer prognostic models were superior to rMRS in predicting outcomes in patients with PSC.

#### Inclusion and Exclusion Criteria

Studies that included both adults and children with PSC or studies that were primarily of large-duct PSC that also included subjects with small-duct PSC were included. Articles were excluded if they were not published in English, the full text was not available, they included only pediatric patients, an invasive test or imaging test was part of the prognostic model, they included biomarkers or tests were not commercially available, or they did not compare rMRS in the analysis.

### 2.2. Data Extraction

The two authors extracted pertinent data on studies that specifically mentioned the rMRS in relation to prognostic models. Primary outcomes measured included overall survival, liver-transplant-free survival, hepatic decompensation, cholangiocarcinoma, or a composite outcome. All relevant information was reviewed independently by the two authors, and pertinent data were collected in a standardized manner that included the association between the test or prognostic model and the primary outcome. In cases of disagreement between the two authors, discrepancies were resolved through discussion.

#### Evaluation of Bias

The assessment of bias for each study was determined using the Prediction model study Risk of Bias ASsessment Tool (PROBAST) [6]. An assessment of risk was assigned for each domain of the tool as low, high, or unclear risk of bias. An overall judgment of risk of bias was assigned to each study.

## 3. Results

Thirty-seven studies were identified, of which 21 were excluded because they did not meet the inclusion/exclusion criteria. Another 11 studies were excluded because they did not analyze rMRS, leaving five studies that collectively included 3230 subjects for the systematic review (Figure 1). The rMRS study included 529 subjects. The prognostic models or biomarkers and outcomes included the Amsterdam–Oxford model, UK-PSC score, enhanced liver fibrosis (ELF) score, and PREsTo score (Table 1). Outcomes included overall survival, transplant-free survival, hepatic decompensation, cholangiocarcinoma, or a composite outcome.

### 3.1. The Mayo Risk Score

The revised Mayo Risk Score was published in 2000 and aimed to eliminate histologic staging and more subjective variables (e.g., inflammatory bowel disease, splenomegaly) to increase the model’s ease of use without compromising its accuracy. The derivation data set included 405 patients while the validation set included 124 patients. The revised model includes age, total bilirubin, AST, variceal bleeding, and albumin. The rMRS estimates survival up to 4 years of follow-up and has comparable accuracy to the original model [7]. The survival function at 4 years for the rMRS was 0.833 (Table 2).

### 3.2. Amsterdam–Oxford Model

The AOM was developed in 692 patients with PSC (5% had PSC–AIH overlap) and validated in 264 patients (2% had PSC–AIH overlap) [8]. The components of the model are shown in Figure 2. The c-statistic was 0.68 (95% CI 0.51–0.85) for PSC-related death and liver transplantation, but was not compared to the rMRS. Goet et al. performed a study to validate the AOM using data from three European tertiary care centers [9]. The study included 534 patients with PSC; 466 (87%) had large-duct PSC, 52 (10%) had overlap with AIH, and 16 (3%) had small-duct PSC. A combined endpoint of liver transplant and all-cause mortality was assessed and the AOM was calculated on a yearly basis with a median follow-up of 7.8 years. The c-statistic ranged from 0.67 at baseline to 0.75 at 5 years of follow-up (Table 2).

In a subgroup of patients who had rMRS calculated at baseline (*n* = 498) and at 1 year (*n* = 482) following diagnosis, the rMRS had higher c-statistics than the AOM at baseline (0.73 vs. 0.68) and at 1 year (0.75 vs. 0.70). However, based on comparison with Kaplan–Meier estimates, the rMRS overestimated the risk of liver transplant or death by 5.1% at 1 year, 6.9% at 2 years, 8.9% at 3 years, and 9.6% at 4 years of follow-up [9].

### 3.3. Enhanced Liver Fibrosis

ELF is a non-invasive test that measures three circulating markers of hepatic matrix metabolism: hyaluronic acid, tissue inhibitor of metalloproteinases-1, and propeptide of type III procollagen. Of note, these markers are expressed during the early stages of hepatic collagen deposition

A Norwegian study evaluated the ELF score to predict transplant-free survival in PSC. Serum samples were obtained from 305 patients with large-duct PSC, 96 patients with ulcerative colitis (UC), and 100 healthy controls [10]. There were separate derivation and validation cohorts. ELF scores were higher in PSC than in UC and healthy controls. Higher ELF scores were associated with shorter survival with an optimal cutoff of 10.6. The c-statistic showed good discrimination (AUC 0.78–0.79) between PSC patients who died or had a liver transplant after 4 or 10 years of follow-up in both the derivation and validation panels. Multivariate analysis showed that the ELF score and rMRS were independently associated with transplant-free survival. Furthermore, the combined c-statistics for the ELF score and rMRS was 0.81–0.82, demonstrating complementary prognostic value [10].

A retrospective study examined ELF scores in 32 patients with PSC and cholangiocarcinoma (CCA), 36 patients with CCA, and 119 patients with PSC alone [11]. ELF scores were significantly higher in CCA without chronic liver disease and in PSC + CCA compared to PSC alone (*p* < 0.001). By multivariate analysis, MRS (OR 8.14), CCA (OR 5.36), and older age (OR 1.07) were associated with an elevated ELF score. An optimal cutoff ELF score of 10.1 was associated with a c-statistic of 0.74, a sensitivity of 81%, and a specificity of 60% [11].

### 3.4. UK-PSC Score

Data from 1,001 patients with PSC from 108 hospitals and 7 transplant centers were analyzed to develop a prognostic model for short- and long-term transplant-free survival [12].

Two models were created; a short-term risk score (RS_ST_) used four variables (bilirubin, albumin, hemoglobin, and platelet count) at t_0_ to predict the 2-year outcome (Figure 2). A long-term risk score (RS_LT_) incorporated seven variables obtained at diagnosis or 2 years later to predict the 10-year outcome: age at diagnosis, bilirubin at t_2_, AP at t_2_, platelets at t_2_, presence of extrahepatic biliary disease at t_0_, and variceal hemorrhage by t_2_. The predictive accuracy of the RS_ST_ and RS_LT_ were compared to the rMRS. The UK-PSC score outperformed the rMRS in both the derivation (c-statistic 0.81 for R_ST_ vs. 0.75 for rMRS and 0.80 for R_LT_ vs. 0.79 for rMRS) and validation cohorts (c-statistic 0.81 for R_ST_ vs. 0.73 for rMRS and 0.85 for R_LT_ vs. 0.69 for rMRS) [12] (Table 2).

### 3.5. Primary Sclerosing Cholangitis Risk Estimate Tool (PREsTo)

More recently, a predictive tool using machine learning was developed. A total of 787 patients were included in the derivation cohort with a median follow-up of 6 years [13]. The validation cohort totaled 278 patients from centers in North America and Norway with a median follow-up of 4.21 years. Of note, patients with small-duct PSC, AIH overlap, or advanced fibrosis (MELD > 14 or portal hypertension) were excluded. The tool consists of nine variables: bilirubin, albumin, AP times the ULN, platelets, AST, hemoglobin, sodium, patient age, and the number of years since PSC was diagnosed. The primary outcome was hepatic decompensation (variceal bleeding, ascites, or hepatic encephalopathy).

PREsTo accurately predicted the 5-year probability of decompensation in the derivation (c-statistic 0.96) and validation (c-statistic 0.90) cohorts. In the validation cohort, PREsTo compared favorably to the rMRS (c-statistic 0.85), MELD score (c-statistic 0.85), and AP < 1.5 ULN (c-statistic 0.65). The accuracy of PREsTo was maintained when evaluating the secondary composite endpoint of decompensation, transplant for nonmalignant condition, or death from PSC-related cause (excluding malignancy); c-statistic 0.89 in the derivation and 0.76 in the validation cohort. A subgroup of 114 patients had ELF scores obtained within 3 months of baseline PREsTo score, demonstrating reasonable test performance (c-statistic 0.75). Exploratory analyses did not show any added benefit of adding liver stiffness measured by MRE or fibrosis stage from liver biopsy to PREsTo [13].

#### Risk of Bias

Risk of bias as assessed by the PROBAST tool was low for the majority of studies that included derivation and validation cohorts (Figure 3). The risk of bias was assessed as high when a validation cohort was not included or if the investigators were not blinded to the study outcome.

## 4. Discussion

Prognostic models for PSC have evolved over time, though further modifications will be needed before they are adopted into everyday clinical practice or drug trials. The heterogeneous nature of PSC along with the typically slow progression from diagnosis to clinically relevant endpoints complicate the design of clinical models. Compounding the problem, existing data are based on different study populations, primary endpoints, and durations of follow-up. These differences have limited the ability to directly compare the accuracy of prognostic models. 

Assessing the reduction in alkaline phosphatase (ALP) levels has been of interest due to its simplicity, ready availability, and association with outcomes in primary biliary cholangitis (PBC) [14,15,16]. Varying definitions of ALP response (e.g., <1.5 ULN vs. normalization) have been used in PSC. ALP improvement has been associated with improved transplant-free survival independent of the Rmrs [15,16]. Notably, ALP is not a component of the original or rMRS [7,14]. More direct comparisons between ALP and rMRS have not been well studied.

The original Mayo Risk Score included histologic stage [14]. The model was limited because liver biopsy is invasive and there is potential for sampling error, especially in a disease such as PSC where fibrosis can be patchy. The MRS was revised to eliminate the need for liver biopsy and includes less subjective variables. There are some limitations of the rMRS, which has prompted further research. Three of the five variables (bilirubin, variceal bleeding, albumin) are associated with advanced fibrosis and only a 4-year mortality risk is calculated. The model only estimates time to death from all causes and not time to liver transplant. For those patients that underwent liver transplant, death was projected by how long they would have lived in the absence of transplant. An ideal model would predict risk of mortality, liver transplant, and other clinical endpoints over a longer duration in patients with early-stage disease. The clinical relevance of the rMRS was called into question when a study of high-dose ursodeoxycholic acid showed that rMRS did not predict poor clinical outcomes in the treatment group [17].

There are limited data that compare biomarkers and other predictive models to the rMRS. There has been interest in the ELF score since it measures markers of fibrosis and is a non-invasive test. There was a modest improvement in the AUC (0.78–0.79 to 0.81–0.82) measuring transplant-free survival when the rMRS was used in combination with ELF. ELF, however, is proprietary, is associated with additional costs that are not incurred with other prognostic models, it is not available in some countries, and its application in everyday practice may not be feasible.

This systematic review investigated three other models (AOM, UK-PSC, and PREsTo) and how they compare with the rMRS. The rMRS outperformed the AOM in terms of c-statistics, though neither model exceeded a c-statistic of 0.8, the threshold typically considered a strong prognostic model. Goet et al. demonstrated that the rMRS overestimated the risk of liver transplant or death based on Kaplan–Meier estimates [9]. This may reflect the presence of more advanced PSC cases in their study compared to the population in the AOM study. The UK-PSC study provided a longer time horizon of 10 years with a good c-statistic (>0.8) [12]. Finally, the PREsTo model is unique in that it was constructed using machine learning [13]. In contrast to the UK-PSC study, the PREsTo study excluded those with AIH overlap, small-duct PSC, and advanced PSC based on MELD or the presence of portal hypertension. The primary endpoint (hepatic decompensation) differed from other models. Regardless, the performance characteristics were good, with c-statistics > 0.8.

While the PREsTo model did not demonstrate improved accuracy when fibrosis by MRE was incorporated, there remains an interest in using MR techniques to non-invasively stage fibrosis and characterize cholangiographic findings over time [18,19]. Future studies in this area will need standardization of the study population, clinical endpoints, and duration of follow-up. Although histology is rarely used now, the addition of non-invasive fibrosis and cholangiography measurements may refine existing and future predictive models.

The study by de Vries et al. of the Amsterdam–Oxford model (AOM) for PSC included a derivation and validation cohort, and a separate retrospective study demonstrated the utility of the AOM to predict PSC-related death and liver transplant. Readily available variables were included in the model. Of note, small numbers of small-duct PSC patients (~10%) were included. The discriminative power of the model was deemed satisfactory with a c-statistic of 0.68 in the validation cohort. The authors could not directly compare their model to the rMRS because of the lack of data regarding a history of variceal bleeding and this study was excluded. However, the study by Goet et al. compared the AOM to rMRS and had a better test performance compared to AOM [9].

There were several limitations to our systematic review. We only included studies published in English and as full manuscripts. Abstracts were not included because of the lack of information to assess risk of bias. The studies of AOM included a small number of patients with AIH–PSC overlap, while other studies excluded these patients. We intentionally excluded studies of biomarkers that are not readily available, such as autotaxin or miRNAs, because they cannot be routinely applied in clinical practice [20,21]. Studies of MRI, MRCP, or elastography were excluded because they are not available as point-of-care tests nor have they been definitively shown to be better than prognostic models that include clinical variables and blood-based tests. However, imaging studies and elastography show promise and warrant further study.

## 5. Conclusions

In conclusion, the UK-PSC score and PREsTo are modestly better at predicting clinical outcomes compared to rMRS. However, studies use different clinical outcomes and variable time horizons to assess model performance. Moving forward, the same clinical outcome and time horizon should be used across studies and newer models should be compared to the rMRS.

## Figures and Tables

**Figure 1 jcm-10-04476-f001:**
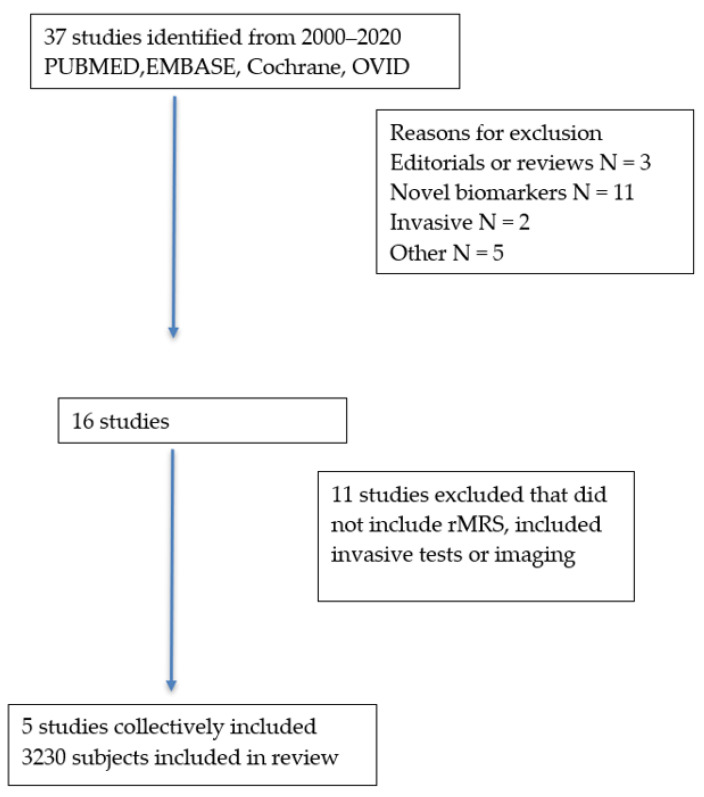
Selection of studies.

**Figure 2 jcm-10-04476-f002:**
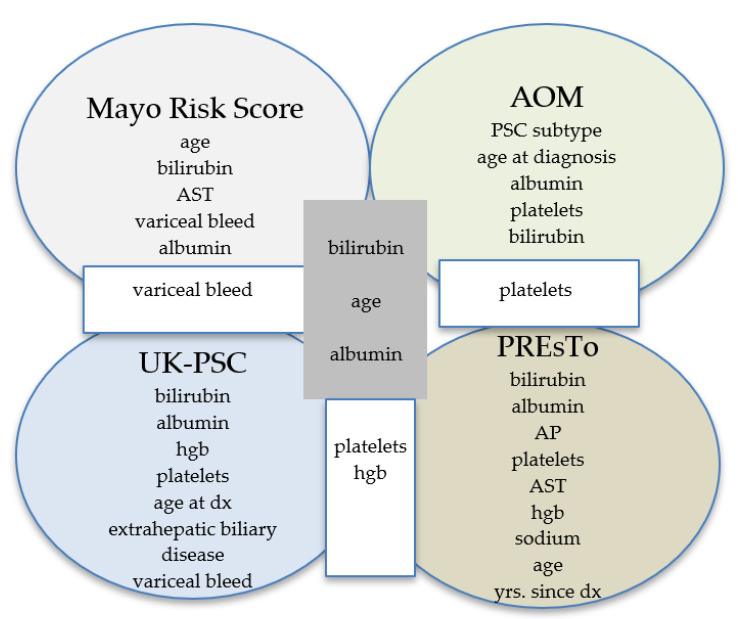
Comparison of components of prognostic models. AOM = Amsterdam–Oxford model, PREsTo = Primary Sclerosing Cholangitis Risk Estimate Tool.

**Figure 3 jcm-10-04476-f003:**
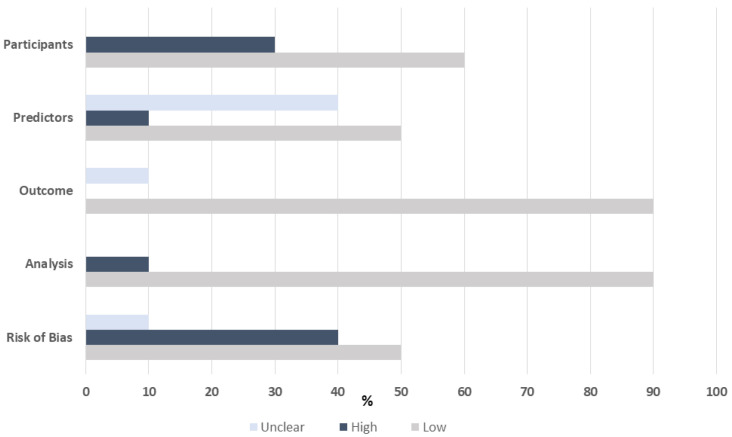
Risk of bias assessed by PROBAST tool.

**Table 1 jcm-10-04476-t001:** Description of studies.

Study	Design	Outcomes
Kim et al. rMRS	Retrospective Derivation and validation cohorts	4 years survival Low risk ~90% Intermediate risk ~60% High risk ~20%
Amsterdam–Oxford model (9)	Retrospective	Transplant-free survival Low risk vs. high group 62% vs. 95%
ELF score ELF test (10) ELF score (11)	Retrospective Derivation and validation cohorts	Death or liver transplant ELF score HR = 1.51 (95% CI 1.06–2.14) ELF test HR = 1.46 (95% CI 1.13–1.88) MRS HR = 1.58 (95% CI 1.08–1.18–2.12) Median TFS by ELF score tertile Lowest 10 years Intermediate 7.5 years Highest 0.8 years *p* < 0.001 Cholangiocarcinoma ELF > 9.8 aOR = 4.91 (95% CI 1.19–20.21)
UK-PSC score (12)	Retrospective Derivation and two validation cohorts	10-year rate of liver transplant Low UK-PSC score 2.9% High UK-PSC score 47.9%.
PREsTo score (13)	Retrospective Derivation and validation cohorts	Hepatic decompensation MRE + PREsTo c-statistic = 0.94

**Table 2 jcm-10-04476-t002:** Comparison of PSC prognostic models with rMRS.

Model	Primary Endpoint	Time Horizon	rMRS c-Statistic	Model c-Statistic
Revised Mayo Risk Score	Overall survival	4 years	NR	NA
Amsterdam–Oxford model	Transplant-free survival	1 year	0.75	0.70
ELF	Death or liver transplant	4 years 10 years	0.81 for MRS + ELF score	ELF score 0.78 @ 10 years 0.79 @ 4 years
UK-PSC score	Transplant-free survival	2 years 10 years	0.73 0.69	0.81 0.85
PREsTo	Hepatic decompensation	5 years	0.85	0.90

No association between rMRS and cholangiocarcinoma (CCA) by multivariate analysis.

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
