# Peer review of "Systematic Review of Prognostic Models Compared to the Mayo Risk Score for Primary Sclerosing Cholangitis"

_jcm, 2021, doi:10.3390/jcm10194476_

Round 1

Reviewer 1 Report

Primary sclerosing cholangitis (PSC) is a slowly progressive biliary disease, which affects predominantly young to middle-aged adults and leads to end-stage liver disease in a significant proportion of cases. Moreover, the disease is associated with the risk of the occurrence of cholangiocarcinoma, a malignancy with a disastrous prognosis. Therefore, risk stratification is crucial for making clinical decisions, however it is hindered by a scarcity of proven prognostic markers. Novel prediction models for the risk stratification in PSC are therefore urgently needed and being actively pursued. The first non-invasive model, namely Mayo Risk Score for PSC, has been developed in 2000 to predict hepatic decompensation or poor transplantation-free survival. Recently, new scoring systems, representing a combination of simple biochemical and clinical variables, were proposed as novel non-invasive tools for outcome prediction, however their utility weren’t systematically compared up to date. The authors aimed to review the literature for newer non-invasive prognostic models for PSC, incorporating routinely available clinical and laboratory parameters, and to compare their utility to revised Mayo Risk Score. The authors included into the systematic review 5 studies (n=3230) on 4 models: Amsterdam-Oxford model, UK-PSC score, enhanced liver fibrosis (ELF) score, and PREsTo score. A PROBAST tool was applied for the estimation of the risk of bias. The comparison of new prognostic models to revised Mayo Risk Score was based on the presentation of c-statistic. The authors showed that the UK-PSC score and PREsTo were modestly better at predicting clinical outcomes compared to revised Mayo Risk Score. The work by P. Schmeltzer and M. Russo is current and of key clinical importance. The manuscript is well written and methodically correct. The conclusions drawn from the review may help clinicians to choose an optimal model for surveillance for their patients. However, the comparative analysis is hindered by the fundamental differences between the studies in regard of primary outcomes measures and length of the follow-up. The authors pointed this out as a limitation of the study and indicate the need for new directions for further studies on the outcome prediction in patients with PSC. Minor points There are several typing errors, the entire manuscript should be carefully checked before publication. Page 7: Please change the word “ursodiol” for “ursodeoxycholic acid”.

Author Response

We thank the reviewer for taking the time to review our manuscript and provide a point-by-point response below.

Primary sclerosing cholangitis (PSC) is a slowly progressive biliary disease, which affects predominantly young to middle-aged adults and leads to end-stage liver disease in a significant proportion of cases. Moreover, the disease is associated with the risk of the occurrence of cholangiocarcinoma, a malignancy with a disastrous prognosis. Therefore, risk stratification is crucial for making clinical decisions, however it is hindered by a scarcity of proven prognostic markers. Novel prediction models for the risk stratification in PSC are therefore urgently needed and being actively pursued. The first non-invasive model, namely Mayo Risk Score for PSC, has been developed in 2000 to predict hepatic decompensation or poor transplantation-free survival. Recently, new scoring systems, representing a combination of simple biochemical and clinical variables, were proposed as novel non-invasive tools for outcome prediction, however their utility weren’t systematically compared up to date. The authors aimed to review the literature for newer non-invasive prognostic models for PSC, incorporating routinely available clinical and laboratory parameters, and to compare their utility to revised Mayo Risk Score. The authors included into the systematic review 5 studies (n=3230) on 4 models: Amsterdam-Oxford model, UK-PSC score, enhanced liver fibrosis (ELF) score, and PREsTo score. A PROBAST tool was applied for the estimation of the risk of bias. The comparison of new prognostic models to revised Mayo Risk Score was based on the presentation of c-statistic. The authors showed that the UK-PSC score and PREsTo were modestly better at predicting clinical outcomes compared to revised Mayo Risk Score. The work by P. Schmeltzer and M. Russo is current and of key clinical importance.

The manuscript is well written and methodically correct. The conclusions drawn from the review may help clinicians to choose an optimal model for surveillance for their patients. However, the comparative analysis is hindered by the fundamental differences between the studies in regard of primary outcomes measures and length of the follow-up. The authors pointed this out as a limitation of the study and indicate the need for new directions for further studies on the outcome prediction in patients with PSC. Minor points There are several typing errors, the entire manuscript should be carefully checked before publication. Page 7: Please change the word “ursodiol” for “ursodeoxycholic acid”.

RESPONSE: We appreciate the reviewers complimentary comments on our systematic review. We have thoroughly reviewed the manuscript and corrected typing errors. We changed ursodiol to urodeoxycholic acid.

Reviewer 2 Report

The present systematic review examined the accuracy of several possible prognostic models at predicting any clinical endpoints in PSC. 

They investigated these prognostic scores (each of them sharing a few laboratory parameters) coming from a few (n=5) retrospective studies, including a total of 3,230 subjects, validating these scores on heterogeneous sample sizes populations, and producing diverse primary endpoints (e.g., overall survival, transplant-free survival, mortality or liver transplantation, or hepatic decompensation) from each other and having different time intervals (ranging from 1 to 10 years). 

This study is well-written and interesting since it could have the potential to be useful in daily clinical practice, particularly for hepatologists, but because of the not negligible risk of bias of this study and small sample size of examined patients, overall the results are still immature to give strong recommendations. 

However, the authors did not consider in their inclusion criteria the possibility of PSC patients overlapping with AIH, as it occurs in the analyzed study of Goet et al.; otherwise, the same patients were excluded in the study of Eaton JE et al.  In my opinion, the overlap with AIH should be deemed an exclusion criterion to decrease the heterogeneity of the considered papers. 

Additionally, amongst the investigated scores, the ELF score may be applied only in cases of extremely specialized confined centers, while its application in everyday practice in different hepatology/gastroenterology centers does not seem feasible.

Consequently, I believe it would be more reasonable and appropriate to turn the paper into a review study. 

Author Response

We thank the reviewer for taking the time to review our manuscript and provide a point-by-point response below.

The present systematic review examined the accuracy of several possible prognostic models at predicting any clinical endpoints in PSC. 

They investigated these prognostic scores (each of them sharing a few laboratory parameters) coming from a few (n=5) retrospective studies, including a total of 3,230 subjects, validating these scores on heterogeneous sample sizes populations, and producing diverse primary endpoints (e.g., overall survival, transplant-free survival, mortality or liver transplantation, or hepatic decompensation) from each other and having different time intervals (ranging from 1 to 10 years). 

This study is well-written and interesting since it could have the potential to be useful in daily clinical practice, particularly for hepatologists, but because of the not negligible risk of bias of this study and small sample size of examined patients, overall the results are still immature to give strong recommendations.  However, the authors did not consider in their inclusion criteria the possibility of PSC patients overlapping with AIH, as it occurs in the analyzed study of Goet et al.; otherwise, the same patients were excluded in the study of Eaton JE et al.  In my opinion, the overlap with AIH should be deemed an exclusion criterion to decrease the heterogeneity of the considered papers. 

RESPONSE: We appreciate the reviewer’s comments on our manuscript. Although the studies by Goet, et al. and de Vries, et al. (both studies of the Amsterdam-Oxford model) included patients with PSC-AIH overlap, as stated on page 4 of the revised manuscript, they constituted on 10% and 2%-5% of the study population. Therefore, because the readers are made aware of the inclusion of these subjects and they constituted only a small fraction of the entire study population we believe these studies of the AOM model should be included. In addition, in page 8 of the Discussion we mention this as a limitation.

Additionally, amongst the investigated scores, the ELF score may be applied only in cases of extremely specialized confined centers, while its application in everyday practice in different hepatology/gastroenterology centers does not seem feasible.

RESPONSE: In the Discussion of the revised manuscript on page 7 we add the limitation raised by the reviewer that application of ELF in every day practice may not be feasible.

Consequently, I believe it would be more reasonable and appropriate to turn the paper into a review study. 

RESPONSE: We respectfully disagree with the reviewer. As stated in the Methods, we applied robust criteria described in the MOOSE guidelines. Data extraction was conducted in a standardized manner. Evaluation of bias for each study was assessed with the PROBAST tool. Thus, our manuscript applied stringent methods that are not typically applied in a standard review.

Round 2

Reviewer 2 Report

The paper is better organized in its current form.